# High Dynamic Range Imaging with TDC-Based CMOS SPAD Arrays

**Majid Zarghami [1,2,\*], Leonardo Gasparini [1], Matteo Perenzoni [1]**  **and Lucio Pancheri [1,2]**

[1]   Fondazione Bruno Kessler (FBK), Center for Material and Microsystems (CMM), 38123 Trento, Italy
[2]   Department of Industrial Engineering, University of Trento, 38123 Trento, Italy
\*   Correspondence: zarghami@fbk.eu; Tel.: +39-339-564-8984

**Abstract:** This paper investigates the use of image sensors based on complementary metal–oxide–semiconductor (CMOS) single-photon avalanche diodes (SPADs) in high dynamic range (HDR) imaging by combining photon counts and timestamps. The proposed method is validated experimentally with an SPAD detector based on a per-pixel time-to-digital converter (TDC) architecture. The detector, featuring 32 × 32 pixels with 44.64-μm pitch, 19.48% fill factor, and time-resolving capability of ~295-ps, was fabricated in a 150-nm CMOS standard technology. At high photon flux densities, the pixel output is saturated when operating in photon-counting mode, thus limiting the DR of this imager. This limitation can be overcome by exploiting the distribution of photon arrival times in each pixel, which shows an exponential behavior with a decay rate dependent on the photon flux level. By fitting the histogram curve with the exponential decay function, the extracted time constant is used to estimate the photon count. This approach achieves 138.7-dB dynamic range within 30-ms of integration time, and can be further extended by using a timestamping mechanism with a higher resolution.

**Keywords:** single-photon avalanche diode (SPAD); high dynamic range (HDR); imager; photon counting; time-to-digital converter (TDC); photon arrival time

---

## 1. Introduction

Applications such as security surveillance, astronomy, automotive, and scientific imaging demand detectors with single-photon sensitivity to achieve a high signal-to-noise ratio (SNR) in all light conditions and with a high dynamic range (HDR) to cope with large variations of the background [1–5]. For example, in automotive applications, a highly demanding condition occurs when entering a tunnel on a sunny day. Two basic scenarios are typically adopted to extend the DR of conventional pixels, either operating in the linear mode, or applying some compressive function [6–9].

A range of pixel technologies are currently available with single-photon sensitivity, for instance electron-multiplying charge-coupled device (EMCCD), photo-cathode-based intensified CCD (ICCD), and intensified complementary metal–oxide–semiconductor (ICMOS). These technologies, however, present some limitations; EMCCDs require a cooling system, ICCDs are costly, and all these technologies can tune the gain to offer either high single-photon sensitivity or HDR but not both [10]. Another pixel paradigm achieving single-photon detection capability is the quanta image sensor (QIS), which avoids the need to use avalanche multiplication. This technology allows low dark count rates, smaller pixel pitch, and also achieves an HDR by merging the information gathered from multiple pixels (jots) and/or multiple exposures [11,12].

CMOS SPAD image sensors digitize individual photons into digital pulses, and these pulses are then processed either to count photons [13] (to quantify the light intensity of the scene) or to sample the photons arrival times with ~100-ps timing resolution [14]. CMOS SPAD pixels as photon counter devices

have the intrinsic capability of capturing HDR scenes (above 100 dB) [15–17]; in theory, each SPAD can cover the range from 1 count/s, owing to its single photon sensitivity, to several million counts/s. However, in practice, the minimum detectable count is determined by the dark count rate (DCR), and the maximum count is limited by the SPAD dead time, which depends on the implemented quenching mechanism [18]. The SPAD pulse counting mechanism is implemented with three different methods: using digital counters, analog counters, or 1-bit memories [10]. Digital counters are area-consuming and, thus, require a very advanced technology node or three-dimensional (3D) stacking. For example, a pixel with 10-bit depth and 20% fill factor cannot fit on a pitch smaller than 44 μm in a 150-nm technology node. Analog counting is implemented in two different methods, switched current source and charge transfer, which are promising in terms of area; however, so far, no implementation in the literature achieved more than 6 bits [19]. In 1-bit memory structures, the state of the memory is switched as a consequence of the SPAD event (photon or dark) within a time-gated exposure window following the quanta image sensor (QIS) concept. The binary maps generated by the array require a high frame rate and external storage to build up an image. The timestamping of the incident photon (known as spatiotemporal image) is conducted in two domains: time-to-analog converters (TACs) and time-to-digital converters (TDCs). TACs have good performance in terms of low power consumption and compactness, but need a highly parallel, integrated analog-to-digital conversion stage with offset compensation in order to achieve reasonably high frame rates and compensate non-uniformity. These features are improved in TDCs, as they are intrinsically digital; on the contrary, TDC implementations have low fill factor and may suffer from high power consumption [20,21].

The main motivation of this work is, thus, to study experimentally how the arrival time of incident photons can be employed to acquire an HDR scene using an imager that combines an SPAD and a TDC in each pixel. The flux of photons from most light sources is well described by a Poisson process. Therefore, the average number of photon arrivals over a time interval depends only on the average arrival rate and the length of the interval. According to Poisson statistics, the timestamp of a photon detected after a time *t* from an arbitrary starting time has an exponential probability distribution [22] that depends on the light intensity. We propose a new method for HDR imaging that exploits the great time-resolving capability of SPADs to estimate the photon arrival time information and, thus, overcome the maximum count limit. With such a method, which can be classified as a compression technique, the average photon arrival rate is reconstructed from the histogram of the generated TDC codes. Hence, the estimated photon count is affected by the TDC characteristics, namely, the TDC's resolution, nonlinearities, quantization error, offset, and gain error.

The paper is organized as follows: Section 2 describes briefly the SPAD imager with its operation. Section 3 outlines how the proposed method creates an HDR scene with a TDC-based SPAD imager. Experimental results in Section 4 demonstrate proof of the concept. Section 5 discusses the obtained results and proposes a possible future implementation.

## 2. SuperEllen Chip

SuperEllen is a time-resolved single-photon imager based on SPADs implemented in 150-nm CMOS standard technology. It was specifically designed for quantum imaging applications for the detection of entangled photon states using a per-pixel TDC configuration [23]. The design of the SuperEllen chip with its characterization is described in Reference [24]. The sensor consists of a $32 \times 32$ array of pixels of 44.64-μm pitch, for a total size of $1.4 \times 1.4$ mm$^2$, achieving a notable pixel fill factor of 19.48%. The TDC resolution can be tuned in the 200–300-ps range by operating on a reference voltage $V_{reg}$. This sensor can acquire frames up to a rate of 800 kfps at low photon flux (dark condition) thanks to zero-suppression strategies implemented on-chip that reduce the readout time (and the power consumption) with a negligible impact on the fill factor. Figure 1a shows the pixel block diagram containing an SPAD with its biasing circuitry, a gating front-end, an 8-bit TDC, and a few high-speed readout components. A dedicated programmable 1-bit memory cell in each pixel enables and disables the SPAD. This is mostly useful to keep the high-DCR SPADs off and consequently

reduce cross-talk events. Photon detection probability (PDP), DCR, and cross-talk can be traded off operating on the SPAD excess bias voltage $V_{ex}$. With the main objective of reducing the DCR level in this experiment, $V_{ex}$ was set ~1.3-V, for which the median DCR is ~240-cps at room temperature, and pixel-to-pixel cross-talk is below $1.5 \times 10^{-5}$ while retaining a peak PDP above 10%, based on the characterization result of similar SPADs [25]. The SPADs are synchronously activated through a global signal, CHARGE, to extract the typical exponential decay of the photon arrival time distribution. This would not be possible with an asynchronous SPAD operation, either using passive or active quenching. START is a step-like signal, which is set high upon the SPAD firing. STOP is a global signal distributed throughout the array. Each TDC converts the time interval between the rising edges of the START and STOP signals into an 8-bit integer code. The high-speed readout circuits implement two zero-suppression functionalities; one method allows a controller to skip rows in case no pixel is triggered in that row, while another one ignores reading out a full frame when the number of triggered pixels is less than a user-defined value. This latter readout mechanism is particularly efficient in quantum optics applications when looking for *N* photon states [23], so that only frames with a number of detected photons equal or greater than *N* are read out. In this specific application, this feature can be used setting the threshold to 1 to skip empty frames and, hence, achieve a higher observation rate when the optical power reaching the focal plane is extremely low.

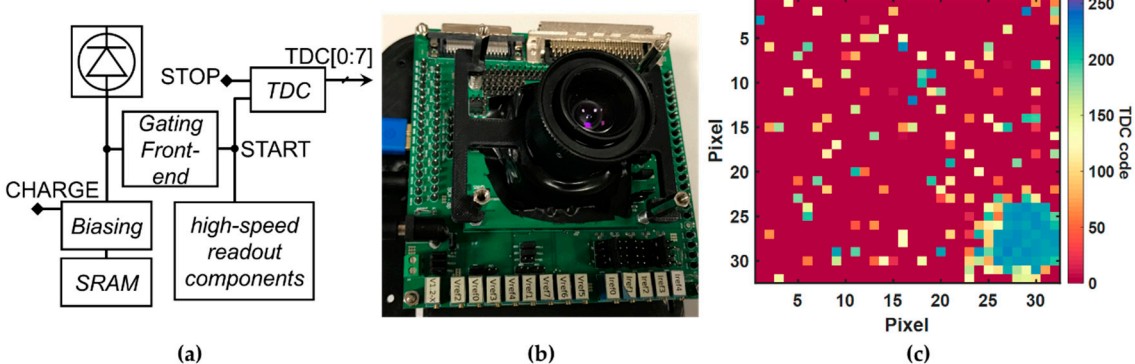

**Figure 1.** (**a**) Pixel block diagram with a single-photon avalanche diode (SPAD), and driving and processing electronics; (**b**) SuperEllen module; (**c**) a sample frame depicting different photon arrival times on the array.

The sensor is mounted on an imaging module which is equipped with an Opal Kelly field-programmable gate array (FPGA) board based on Xilinx Spartan-6 to control and read out the sensor. The module (Figure 1b) is connected via a Universal Serial Bus (USB) 3.0 interface to a computer where a mixed C++/LabVIEW program allows the user to control the sensor and display the acquired data in real time. This software provides a continuous stream of the detected events and their timestamps for consecutive acquisitions. Events belonging to the same acquisition can be represented by a 32 × 32 map visualizing the incident photon timestamps on the array (Figure 1c). In order to build the HDR image by the proposed method, which is fully explained in the next section, we require two pieces of information: the photon counts and the photon arrival time distribution. The per-pixel timestamping capabilities make SuperEllen suitable for studying the HDR properties of an SPAD imager, although, since a counter is not integrated in the pixel, we need to acquire one frame for each count; thus, it is necessary to acquire thousands of frames for a given integration time to generate one HDR image.

## 3. Dynamic Range of SPAD Arrays and Image Formation Model

The stream of photons is typically modeled as a Poisson process for most light sources. Although the stream of photons directed to the detector is subject to various attenuating factors (fill factor, PDP,

filters, and etc.), the detected photons can still be described by the Poisson process. The number of detected photons $n$ within a fixed observation time follows the Poisson distribution given by

$$P(n \, \text{photons per interval}) = \frac{(\overline{n})^n}{n!} e^{-\overline{n}} \qquad n = 0, 1, 2, \ldots, \tag{1}$$

where $\overline{n}$ is the average number of photons per interval and is proportional to the light intensity. A Poisson process is a memory-less process, for which the photon arrival time with respect to any arbitrary reference time point follows the exponential distribution function below.

$$f(t) = \lambda e^{-\lambda t} \qquad t > 0, \tag{2}$$

where $\lambda$ is the arrival rate of the process. This means that the average number of photons detected $\overline{n}$ in an interval $t_{exp}$ is equal to the product $\lambda t_{exp}$. At the same time, $\lambda = 1/\overline{t_{ph}}$, where $\overline{t_{ph}}$ is the average arrival time, such that $\overline{n} = t_{exp}/\overline{t_{ph}}$.

The proposed method is based on dividing the overall exposure time $t_{exp}$ into multiple observation windows (frames) of the same length $t_{frame}$, such that $N_{exp} = t_{exp}/t_{frame}$. Within each observation window, we measure the time of arrival of the first photon $t_{ph}$, which follows an exponential decay distribution. For every exposure, we obtain two values which are used to project $\overline{n}$: (1) the number of frames $N_{ph}$ for which the measured timestamp is non-zero, i.e., the number of frames over $N_{exp}$ frames in which a photon was detected; (2) the average arrival time $\overline{t_{ph}}$. The value that offers the most accurate estimation of $\overline{n}$ can be chosen in different scenarios; $N_{ph}$ can be used when the flux is low, while $\overline{t_{ph}}$ gives a better estimate at high flux densities when $N_{ph}$ saturates to $N_{exp}$.

*3.1. Low Flux*

We count the number of photons $N_{ph}$ detected within the exposure time $t_{exp}$ as mentioned above. The result of this operation is the estimation of the product $\lambda t_{exp}$. To be more precise, $N_{ph}$ is an underestimation of $\lambda t_{exp}$, since the SuperEllen chip can count only one photon for every observation window of size $t_{frame} = 70$-ns. According to Equation (1), for every frame, the probability that we get a non-zero TDC code is given by

$$P(n \geq 1) = \sum_{n=1}^{\infty} P(n) = 1 - P(n = 0) = 1 - e^{-\lambda t_{frame}}. \tag{3}$$

On average, we will then get

$$N_{ph} = \sum_{k=1}^{N_{exp}} \sum_{n=1}^{\infty} 1 \times P(n) = N_{exp} \times P(n \geq 1) = N_{exp} \times \left(1 - e^{-\lambda t_{frame}}\right). \tag{4}$$

This is an underestimate of $\lambda t_{exp}$ which can be calculated by Equation (5).

$$\lambda t_{exp} = \sum_{k=1}^{N_{exp}} \sum_{n=1}^{\infty} n \times P(n). \tag{5}$$

Nevertheless, for $\lambda t_{frame} << 1$, we get $N_{ph} \approx \lambda t_{exp}$. On top of that, we can model the error and use it to compensate our estimate as follows:

$$n_{err} = \lambda t_{exp} - N_{ph} = \sum_{k=1}^{N_{exp}} \sum_{n=2}^{\infty} (n-1) \times P(n). \tag{6}$$

Figure 2 compares the counts measured by an ideal detector and SuperEllen according to the proposed method. The difference is modeled by Equation (6). According to the analytical model, SuperEllen well approximate an ideal detector at low flux densities, while, by increasing the flux densities, the error in photon counting becomes non-negligible at very high flux densities.

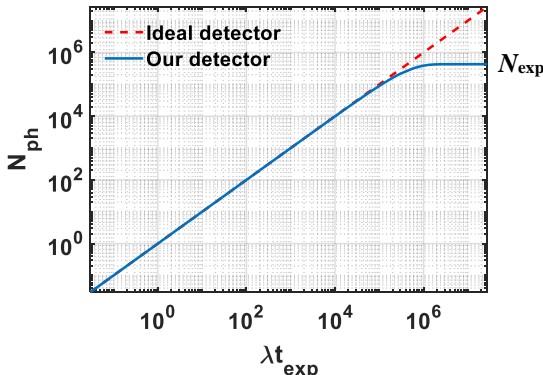

**Figure 2.** Modeling the photon count loss due to the detection of one count per observation window ($t_{exp}$ = 30-ms).

### 3.2. High Flux

According to the modeling result in Figure 2, $N_{ph}$ saturates to $N_{exp}$ at very high flux densities. To extend this upward limit, it would be necessary to reduce the width of the observation window $t_{frame}$, thus increasing the number of accumulated frames $N_{exp}$. However, since the frame readout time is fixed at high flux densities, this solution would result in a lower efficiency and a large reduction of the frame rate. In addition, the attainable range would still be limited by the minimum applicable gating time. Here, we overcome this saturation limit at high flux densities by exploiting the photon arrival time information. The acquired frames form a histogram of the generated TDC codes for every pixel providing the distribution of photon arrival times. The histograms follow an exponential distribution, which is clearly visible for high photon flux densities, and looks linear for lower light intensities (Figure 3c,d)). Data are fitted to a first order exponential decay, $e^{-\lambda t}$ to extract lambda. The photon count $\overline{n}_i$ is then assigned to the pixel for each image $i$, according to the following rule:

$$\overline{n}_i = \begin{cases} N_{ph,i} - N_{dark,i} & 3 \times \lambda_i t_{frame} \leq 1 \\ \lambda_i t_{exp} & 3 \times \lambda_i t_{frame} > 1 \end{cases}, \tag{7}$$

where $N_{dark,i}$ is the number of frames over $N_{exp}$ in which the pixel returns a non-zero TDC code in a dark count experiment. The switching threshold from photon counting mode to exponential fitting mode $3 \times \lambda t_{frame} = 1$ is adequate if the photon count non-linearity is calibrated according to Equation (6). In order to evaluate the HDR performance of our SPAD imager at different frame rates, several $t_{exp}$ are investigated, e.g., in the case of $t_{exp}$ = 210-μs, $N_{exp}$ = 3000 frames are acquired for every image and, thus, the frame rate of 30 images/s can be obtained, while the frame rate drops to 0.2 images/s when $t_{exp}$ = 30-ms. These values can be largely improved by implementing a dedicated architecture.

In practice, each scene is divided into two sub-scenes, having low and high light intensity, and, in each sub-scene, the pixel data are processed in a different way. Each pixel is assigned to one of the two sub-scenes according to the defined boundary condition. For low photon flux densities, the histogram is noisy, and the light intensity is estimated from the number of photons detected over the $N_{exp}$ observations. In this case, a calibration based on the model in Figure 2 is applied to compensate for the lost photons when the flux is close to the switching point for each image. For high photon flux densities, the photon counts saturate to $N_{exp}$, but we can estimate the rate at which the counter saturates by processing the histograms according to the proposed method in Equation (7).

Figure 3 shows histograms of the photon arrival times at three levels of light intensity for a single pixel. The histograms were filtered with a moving average (visualized in Figure 3a,b) to remove the distortion generated by the TDC non-linearity, whose characterization was presented in Reference [24]. The width of the moving average filter was set to four, which corresponds to the periodicity of the differential non-linearity.

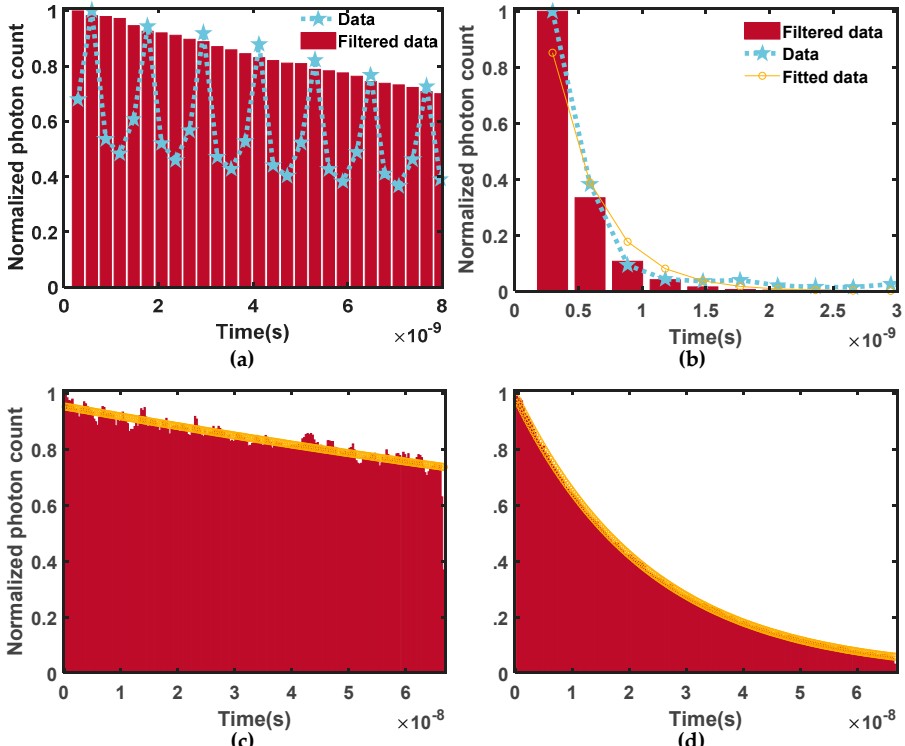

**Figure 3.** Histogram of the photon arrival time at different illumination levels (plots are normalized individually to the corresponding maximum value $t_{exp}$ = 30-ms): (**a**) moving average filters the nonlinearity; (**b**) matching between raw, filtered, and fitted data at high photon flux; (**c,d**) fitted data are aligned to the filtered data at low and medium flux.

The DR for the SPAD imager is defined as follows [13]:

$$DR = 20 \log_{10} \frac{\overline{N}_{\max}}{\overline{N}_{\min}}, \tag{8}$$

where $\overline{N}_{\max}$ is the count estimated by the aforementioned algorithm at very high flux densities, and $\overline{N}_{\min}$ is the photon-induced count that is distinguishable from the noise at very low flux densities. As the sensor is fully digital, it has no read noise; thus, the only contribution to the noise comes from statistical fluctuations of the counts, which at low light intensity are dominated by the shot noise of the dark counts. Moreover, dark counts introduce an offset $N_{dark,i}$ to the measured photon counts in each image which can be removed by means of calibration, as considered in Equation (7), averaging the photon counts acquired in dark conditions over $N_i$ images. This process is effective given that there is no random telegraph signal (RTS) noise affecting that pixel [25]. According to the experimental results, less than 6% of the enabled pixels are affected by RTS noise. The average pixel signal $\overline{N}$ and the pixel noise $\sigma$ are calculated as the mean and standard deviation of the pixel counts over $N_i$ images. In the low flux regime, where the signal fluctuation is well modeled by a Poisson distribution, the standard

deviation is $\sigma = \sqrt{N_{dark}}$. In the end, the minimum detectable signal $\overline{N}_{min}$ is defined as the photon count at the lowest flux density for which the signal-to-noise ratio (SNR), defined as

$$SNR = \frac{\overline{N}}{\sigma}, \tag{9}$$

is equal to one [26].

## 4. Experimental Results

The sensor was tested by shining a white light on the detector at different levels of optical power. Figure 4 shows the experimental set-up used for this characterization, which included a high-intensity fiber light source from Thorlabs (OSL1-EC), a diffuser providing uniform intensity, and several stages of absorptive neutral density filter. In order to guarantee proper control over the input power, the light source was kept at a fixed power, while the beam was attenuated (11 orders of magnitude range) combining multiple filters. The set-up was adjusted so that, at the maximum illumination level, with no filters placed between the sensor and the light source, the average photon arrival time corresponded almost to the TDC resolution, as shown in Figure 3b. In order to characterize the signal and noise, the responses of 49 randomly distributed pixels, choosing a pattern of $7 \times 7$ pixels with a gap of four pixels in between in the $x$- and $y$-directions, were accumulated by acquiring frames for 4 min at each flux level. The measured data did not exhibit large variations of the photon counts over time, confirming the good stability of the photon flux during each measurement. Figure 5a shows the measurement results at different light intensities after removing the DCR offset. The plots include signal $\overline{N}$ and noise (standard deviation of the counts in $N_i$ images) averaged over 49 pixels; their standard deviation, represented with the error bars, describe the pixel uniformity. The blue line shows the extension in dynamic range achieved by estimating the photon counts from the TDC code distribution according to the proposed algorithm. In the low flux region, the nonuniformity is dominated by the pixel-to-pixel variation of DCR. The limitations in the TDC non-linearity correction procedure, including fitting errors and distortions introduced by the moving average when the non-zero bins in the histogram were fewer than the number of averaged bins, yielded a very large noise nonuniformity at high photon flux densities. Figure 5b illustrates the photon transfer curve (PTC), i.e., the noise vs. average signal at different light levels, of three sample pixels. The PTCs were limited by photon shot noise in the low photon counting part, although there were few noisy pixels, which were limited by the shot noise of dark counts. In the high-intensity region, the measured noise was larger than the theoretical shot noise due to detector saturation.

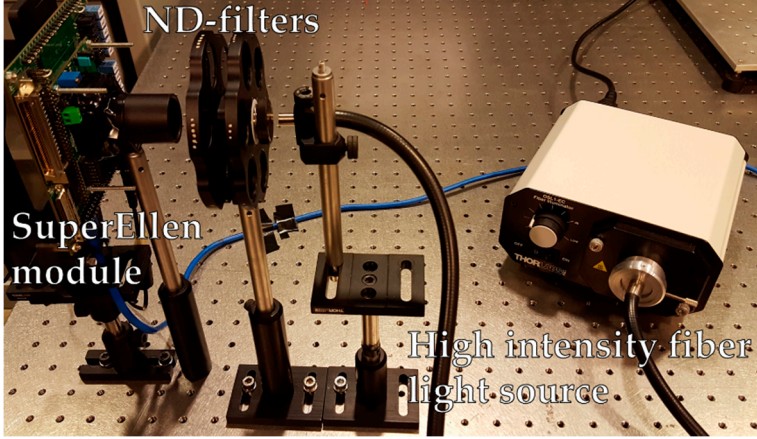

**Figure 4.** Experimental set-up to characterize the sensor dynamic range (DR).

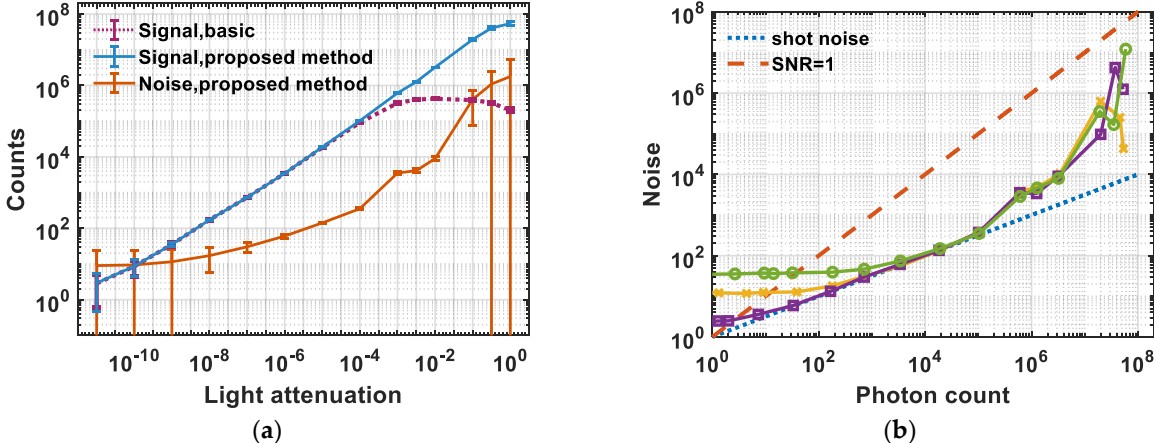

(a)  (b)

**Figure 5.** Both plots are at $t_{exp}$ = 30-ms: (**a**) average and non-uniformity (error bars) of estimated optical signal and noise from 49 pixels; (**b**) photon transfer curve for three pixels.

Figure 6 summarizes the DR and SNR characteristics of SuperEllen by analyzing the same dataset used in Figure 5. Figure 6a shows the DR for different exposure times. The DR was 107-dB with $t_{exp}$ equal to 210-µs and increased up to 138.7-dB when $t_{exp}$ was 30-ms. The first exposure time of 210-µs was reported as it corresponds to a real image rate of 30 images/s. Figure 6b plots the mean SNR and its standard deviation through the error bars calculated over the 49 enabled pixels at different exposure times by using Equation (9). All SNR curves saturated when the photon count was estimated from the TDC code distribution. Intuitively, this was due to the fact that the estimation was based on the number of samples, which was limited upward by the number of frames used to create an image. On top of this, a number of error sources in the photon count estimation increased the noise in the high flux part, including pixel-to-pixel variations of the TDC resolution and its dependence on the number of triggered pixels. In other words, if too many TDCs are running at the same time, the stability of their timing resolution is adversely affected by the ohmic voltage drop in the power distribution lines; hence, we opted to characterize only 49 pixels to guarantee less than 0.035% dB bit error in photon count estimation. Owing to this fact, the design of large TDC arrays operating at high light flux imaging requires careful optimization of power distribution and power consumption. Moreover, the nonlinearity of TDC strongly affects the estimated value of $\lambda$ at very high photon flux densities.

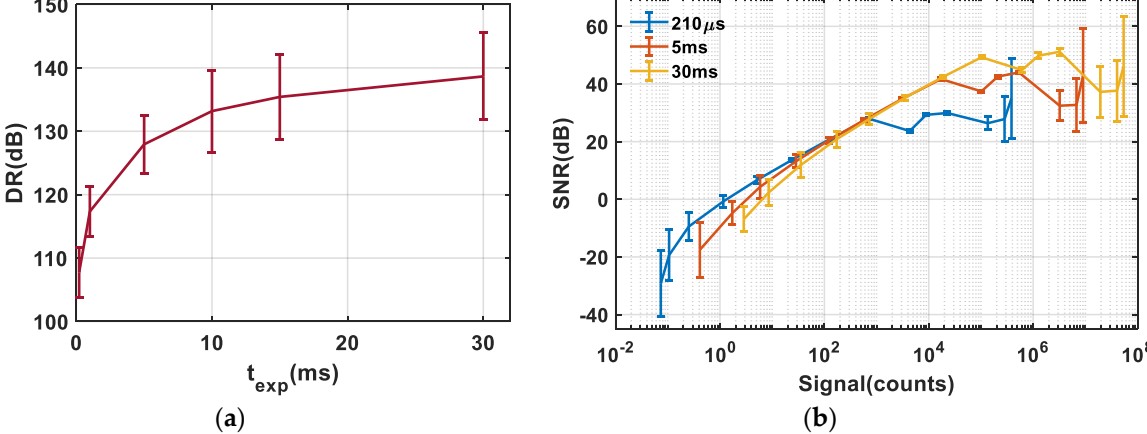

(a)  (b)

**Figure 6.** (**a**) Average DR versus integration time for 49 pixels; (**b**) signal-to-noise ratio (SNR) plot for 49 pixels at 210-µs, 5-ms, and 30-ms of exposure time. Error bars describe the uniformity over the 49 pixels.

Figure 7 demonstrates the sensor HDR operation in a real case representing images of a scene acquired in three modalities: photon counting, timestamping, and using the proposed method. The 32 × 32 SuperEllen sensor was used to capture an HDR scene including a table lamp with an output power of 50 W and a pattern of eight black squares in different contrast starting from 5% to 100% in an indoor environment with black walls and lights off. The sensor was equipped with a low-cost Computar 6-mm f1.2 adjustable lens c-mount to collect the light. The lamp area was the region with very high flux, and the proposed algorithm clearly distinguished different intensities in all the regions of the scene, thus confirming its efficiency. According to the obtained dataset, the entire scene was almost visible, including low-contrast regions, thanks to the SPAD's single-photon sensitivity. However, low-contrast squares (5% and 15%) are not clearly distinguishable in Figure 7d due to the conversion of the HDR image into a color image with a limited number of bits.

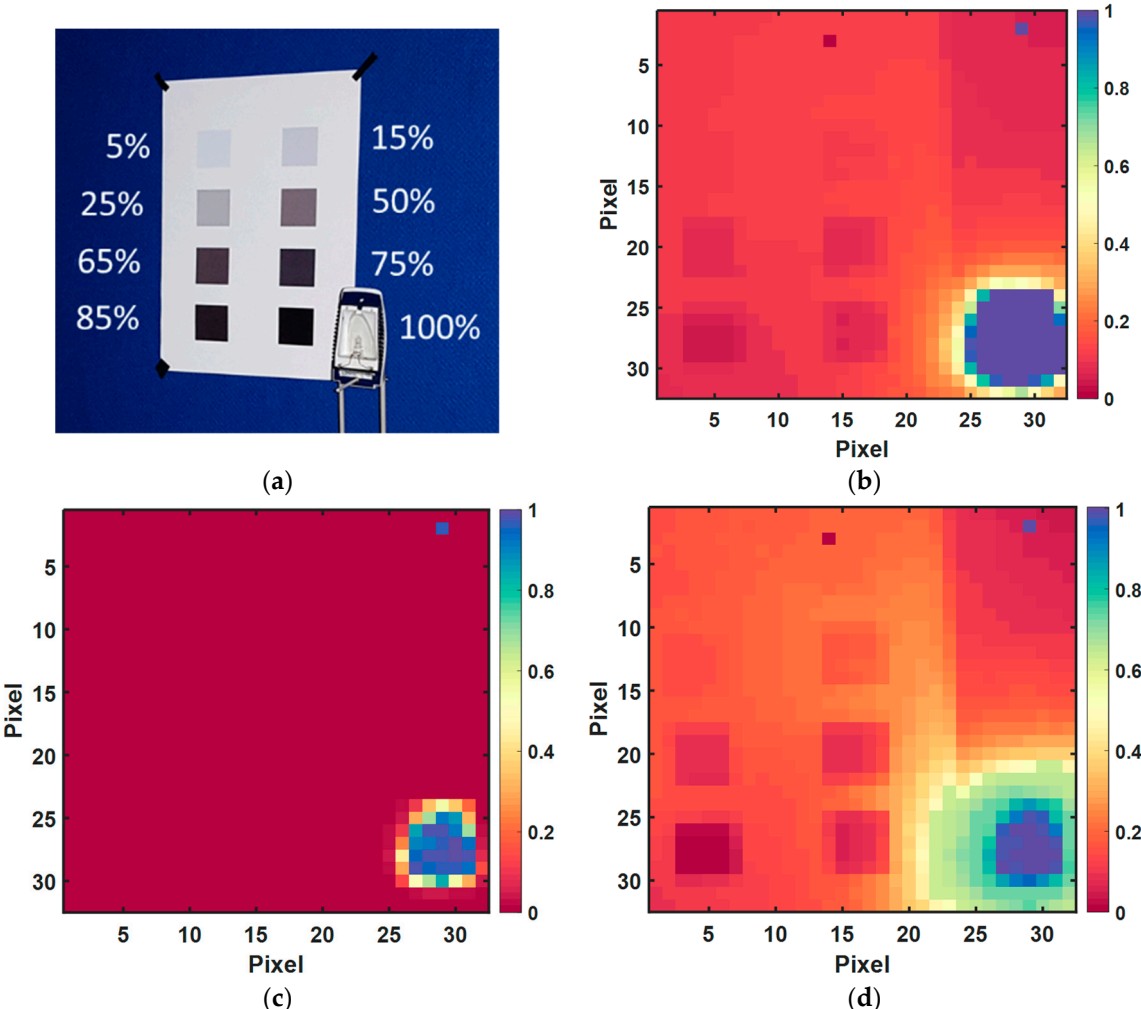

**Figure 7.** Images captured by SuperEllen sensor with 30-ms of exposure time: (**a**) the captured scene, acquired with a conventional camera with typical indoor illumination and the lamp switched off; (**b**) image based on the photon counts; (**c**) image based on the photon arrival times; (**d**) high dynamic range (HDR) image obtained with the proposed method.

## 5. Discussion

In the previous sections, we evaluated the HDR performance of SuperEllen, a CMOS SPAD imager capable of timestamping photons in every pixel by means of a TDC. The photon timestamps were processed to estimate the photon flux when the "standard" photon counting method saturated. Results showed an average DR of 138.7-dB, which exceeded the performance of other single-photon

technologies such as EMCCDs, intensifier-based cameras, and QIS [11]. Table 1 compares the obtained results with the state-of-the-art CMOS SPAD detectors and (CMOS image sensor) CISs. Nevertheless, the employed sensor cannot yet be considered a valid alternative to those technologies, which surpass SPAD imagers with respect to other figures of merit such as pixel pitch, fill factor, and detection efficiency. There is, however, room for improvement both in the pixel design and through the opportunities offered by technology scaling. The sensor was designed for quantum optics application, and its operation can be assumed as a single-bit architecture in our experiment, which requires significant data rate and power to achieve HDR imaging. The sensor readout time is much larger than the 70-ns observation windows adopted in this work, and is responsible for slowing down the real frame rate of the acquisition with respect to the effective one, obtained as the reciprocal of the sum of all the observation windows in an image. For instance, using $t_{exp}$ = 30-ms yields 0.2 images/s in a real scenario. This sub-optimal operation mode, as mentioned above, is due to the incapability of the sensor to count more than one photon in one frame. A promising compact pixel structure allowing the reconstruction of the decay rate could be a multi-time-gated counter. This implementation could be imagined as a follow-up for an imager which was developed for fluorescence lifetime imaging microscopy (FLIM) [19], which would provide data compression due to the on-chip histogramming for each time gate, increasing the on-chip frame rate, and considerably reducing data rates off-chip. In a future implementation, the estimation for the maximum count rate would surpass the limit given by the SPAD dead time as in a TDC implementation, while being more efficient in terms of area and frame rate, possibly approaching the maximum intrinsic limit of the SPAD, which is represented by its jitter.

**Table 1.** Comparison of the high dynamic range (HDR) single-photon avalanche diode (SPAD), (avalanche photodiode) APD, and (CMOS image sensor) CIS imager. TDC—time-to-digital converter; DCR—dark count rate; SNR—signal-to-noise ratio.

| Sensor | This Work | | [27] | [6] | [13] | [16] | [4] |
|---|---|---|---|---|---|---|---|
| Technology (nm) | 150 | | 180 | 350 | 350 | 40/90 | 110 |
| Detector | SPAD 32 × 32 | | SPAD 128 × 1 | photodiode 180 × 148 | SPAD 64 × 32 | SPAD 256 × 256/ 64 × 64 | avalanche photodiode 1280 × 720 |
| Architecture | TDC-based | | Photon counting | | Photon counting | Photon counting | Photon counting |
| Pixel pitch (μm) | 44.64 | | 15 | 33 | 150 | 9.2/38.4 | 3.8 |
| Fill factor | 19.48% | | 4% | 0.8% | 3.14% | 51% | |
| DCR | 240 @ $V_{ex}$ = 1.3 | | 1-k @ $V_{ex}$ = 1.5 | | 100 @ $V_{ex}$ = 5 | 20 @ $V_{ex}$ = 1.5 | 0.1 |
| SNR (dB) | 51.2 | 30 | | | 53.8 | | |
| DR (dB) | 138.7 | 107 | | 80 | 151.45 | 110 | 120 | 100 |
| Frame rate (fps) | 0.2 | 30 | | | 0.125 | 100 | 30 | 15 |
| Integration time (ms) | 30 | 0.21 | | 2 | 8000 | 10 | | 33 |

**Author Contributions:** Conceptualization, L.G. and L.P.; methodology, M.Z., L.G., and L.P.; software, M.Z. and L.G.; validation, M.Z., L.G., and L.P.; formal analysis, M.Z., L.G., and L.P.; investigation, M.Z.; writing—original draft preparation, M.Z. and L.G.; writing—review and editing, M.Z., L.G., and L.P.; visualization, M.Z., L.G., and L.P.; supervision, L.G., M.P., and L.P.

**Funding:** We thankfully acknowledge the support of the European Commission through the SUPERTWIN project, ID 686731.

**Conflicts of Interest:** The authors declare no conflicts of interest.

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
