# Peer review of "High Dynamic Range Imaging with TDC-Based CMOS SPAD Arrays"

_instruments, doi:10.3390/instruments3030038_

Round 1
Reviewer 1 Report
The paper describes a technique to enhance the Dynamic Range of a SPAD imager, exploiting the photon timing information.
The results do not show any evidence that the proposed technique is effective to increase the dynamic range since a fair comparison has not been presented. The theoretical basis, the methods of the measurements and the results have not been properly described, as enlighten by the following comments:
- Low Light and High Dynamic Range in the title seems to be an oxymoron. I suggest to modify it (remove low light, since HDR means both low and high fluxes).
- Sec. 1: it should be explained why linear approaches deteriorates the pixel pitch. CCD or CMOS APS cameras uses linear approaches, with very small pixel pitch.
- Sec. 2: “this is the state of art for TDC-based…” this sentence must be clarified, in particular it must be specified the context within which the array is the state of art (fill-factor? TDC performance? SPAD performance? Frame-rate?). Are Authors sure that considering only one aspect is a fair comparison (e.g., others SPAD arrays can have lower fill-factor, but better TDC resolution or better PDE for instance, thus the two arrays are not comparable).
- Sec. 2: “SPADs are synchronously activated…” does it means that the time interval between the photon detection and a synchronization signal is measured instead of the time interval between two photon detection? This seems to be in disagreement with what was said before.
- Sec. 2: “although, since a counter is not integrated in the pixel, it is necessary to acquire thousands of frames to generate one HDR image”, this sentence is not clear in Sec.2, since it has not been explained yet how the technique is implemented, and it is not clarified in the other sections of the paper.
- Eq. 1: this equation is fundamental for the paper because it summarizes the proposed method. Nevertheless, it is presented as a “rule” without any theoretical justifications. This is probably the main weakness of the paper.
- Sec. 3: The TDC DNL (-0.73 - +0.77 LSB) is quite high, and this is also visible in Fig. 2a. Which is the impact of such DNL. Is the moving average the best way to compensate for DNL? Has the filtered data been normalized to something (it is not intuitive that filtered data almost fit the local maxima of the raw data)
- Fig. 4(a) must be better explained. What is signal and what is noise, how they have been measured. How “signal proposed method” and “noise proposed method” have been extracted from signal? Why only 49 pixels out of 1024 have been considered?
- Fig. 5: how does the DR has been computed? Show the equation. Has the DR been computed using the experimental data of Fig. 4? Has SNR in figure 5b been computed using eq. 2?
- Fig. 6: the text claims that also the regions with lower contrast (thus I suppose 5% and 15%) are visible, instead according to the shown image those regions are not distinguishable.
- Fig. 6: how image 6b and 6c have been extracted is not explained at all. Fig. b can be simply obtained with a counter, but according to sec.2 the counter is not included in the imager. Time-stamp typically is used to reconstruct 3D images, how this information has been used to obtain an intensity information?
- Table 1. The comparison with other detectors is not fair since it has been done at different exposure time. Furthermore, for the other detectors effective exposure time is equal to the inverse of the real frame rate, whereas with the proposed approach the real frame rate is much slower. A fair comparison would be at the same frame rate (and in this case the result achieved with the proposed architecture would be far away worse than the others also in terms of DR).
- Sec. 5: The real frame rate for 5 ms effective exposure is provided, but the frame rate for 30 ms should have been provided, since 30 ms is used through the entire paper.
Other minor comments:
- Sometimes the space between numbers and units is missing.
- Sec. 1: “In the latter case”, what does “latter” refer to?
- Sec. 1: “range from 1count/s to several…” how these lower and upper limits have been established?
- Sec. 1: The meaning of TDC must be explicated.
- Sec. 2: The acronym of Photon Detection Efficiency is PDE (and not PDP).
- Eq. (1) better to use texp,eff instead of 30 ms and tframe instead of 70 ns.
- Fig. 2: swap position of c and d (to better correspondence with a and b)
- Eq. (2) should be better linked to the text or can be removed.
- Sec. 4: briefly describe what the photo transfer curve is.
Reviewer 2 Report
The paper presents good and significant results and is well written. The method is very effective and sound as an engineering method. Some minor points are expected to be improved as follows.
(1) Figure 4 (b): The noise data are not explained such that green, purple, and yellow curves are obtained from 3 samples.
(2) Figure 4 (b): The experimental conditions are expected to be clarified such as exposure time and illumination level (or photon arrival rate).
(3) Reference 19: The volume number or year is expected to be added.
(4) Necessary frame numbers ("thousands of" on line 119, "428572" on line 130) are reasonable considering the level of illumination the authors are challenging. However, if some explanation for these numbers with respect to illumnation condition were made, that would be very helpful to readers.
Reviewer 3 Report
As far as I can see, the basic idea is to measure the radiant exitance of an object by measuring the mean time interval between the detected photons, especially at high illumination level, which could saturate the receiver with standard counting techniques. Instead of directly measuring the inter-photon time interval, the measurement is conducted from the distribution of photon detection moments with respect to a synchronization signal. Since only one TDC channel is in use, the corresponding distribution is exponential and its time constant gives the needed information. The idea is interesting and worth of studying.
More detailed comments:
- the references should, whenever possible, be to the original milestone papers, e.g. with regard to the TDCs.
- the exact application could be described in more details, why the high DR is necessary?
- what is the role of the array, is it needed in the application, or just happened to be there?
- the loading of SPADs takes some time and may produce uncertainty to short time intervals, say <1ns. How did you handle this issue?
- Vex~1.3V is claimed to be optimum, what do you mean by optimum, in what respect?
- the SPAD parameters refer to another paper with many other authors? Is it so that the same device was used there?
- Eq. 1, do the two equations give the same result when tau=70ns? Is it needed? How do you ensure the continuity of the measurement (no step at 70ns)
- the TDC non-linearity seems to be quite large (Fig. 2), probably it is a limitation for these techniques?
- some analysis of noise would be nice to see, how precisely tau can be analyzed in 30ms? For example, in Fig. 2b it is hard to get the tau very precisely (due to the random component of the bin counts)?
- I disagree with you when you say that the only noise source is DCR. Even without DCR and even with an ideal TDC (linear and no jitter) the distributions you measure would have a random component. Please, explain.
- minimum detectable SNR=1 in Eq. 2, how come? Please, explain.
- 4 minutes measurement time versus 30ms integration time? Why 4 minutes?
- why results were averaged over 49 pixels, which 49?
- what is the repeatability for the signal component. Just repeat 30ms measurements and show the measured/calculated intensity and its variation.
- Fig 4b is unclear, what does it mean?
Round 2
Reviewer 1 Report
The paper has been significantly improved and all of the punctual comments have been addressed.
Nevertheless, the most general issue is still open and more evident from Table 1 with the new values.
Which is the advantage of the proposed technique in respect to normal photon counting (i.e. with integrated counter) SPAD arrays?
Indeed [13] and [16] have higher DR at the same or even higher frame rate.
The comparison between figure 6b (photon counting) and 6d (new proposed method) are not enough to demonstrate the validity of the proposed method, since 6b is not obtained with real photon counting, since a counter able to collect many photons per frame is not integrated on-chip.
In my opinion, the main limitation of the suggested method in respect to photon counting is that photon timing arrays, with in-pixel TDC, can detect at most 1 photon per frame, and it is very hard to overcome this limitation, because many memory registers would be needed.
Reviewer 3 Report
Revision OK.